# Renal Tubular TRPA1 as a Risk Factor for Recovery of Renal Function from Acute Tubular Necrosis

**DOI:** 10.3390/jcm8122187

**Published:** 2019-12-11

**Authors:** Chung-Kuan Wu, Chia-Lin Wu, Tzu-Cheng Su, Yu Ru Kou, Chew-Teng Kor, Tzong-Shyuan Lee, Der-Cherng Tarng

**Affiliations:** 1Institute of Clinical Medicine, National Yang-Ming University, Taipei 11221, Taiwan; chungkuan.wu@gmail.com (C.-K.W.); 143843@cch.org.tw (C.-L.W.); 2Division of Nephrology, Department of Internal Medicine, Shin-Kong Wu Ho-Su Memorial Hospital, Taipei 11101, Taiwan; 3School of Medicine, Fu-Jen Catholic University, New Taipei 24205, Taiwan; 4Division of Nephrology, Department of Internal Medicine, Changhua Christian Hospital, Changhua 50006, Taiwan; 5School of Medicine, Chung-Shan Medical University, Taichung 40201, Taiwan; 6Department of Pathology, Changhua Christian Hospital, Changhua 50006, Taiwan; 140062@cch.org.tw; 7Department of Physiology, School of Medicine, National Yang-Ming University, Taipei 11221, Taiwan; yrkou@ym.edu.tw; 8Internal Medicine Research Center, Changhua Christian Hospital, Changhua 50006, Taiwan; 179297@cch.org.tw; 9Department of Physiology, College of Medicine, National Taiwan University, Taipei 10617, Taiwan; ntutslee@ntu.edu.tw; 10Division of Nephrology, Department of Medicine, Taipei Veterans General Hospital, Taipei 11217, Taiwan

**Keywords:** acute kidney injury, acute tubular necrosis, TRPA1, recovery of renal function

## Abstract

Background: Transient receptor potential ankyrin 1 (TRPA1), a redox-sensing Ca^2+^-influx channel, serves as a gatekeeper for inflammation. However, the role of TRPA1 in kidney injury remains elusive. Methods: The retrospective cohort study recruited 46 adult patients with acute kidney injury (AKI) and biopsy-proven acute tubular necrosis (ATN) and followed them up for more than three months. The subjects were divided into high- and low-renal-tubular-TRPA1-expression groups for the comparison of the total recovery of renal function and mortality within three months. The significance of TRPA1 in patient prognosis was evaluated using Kaplan–Meier curves and logistic regression analysis. Results: Of the 46 adult AKI patients with ATN, 12 totally recovered renal function. The expression level of tubular TRPA1 was detected by quantitative analysis of the immunohistochemistry of biopsy specimens from ATN patients. The AKI patients with high tubular TRPA1 expression showed a high incidence of nontotal renal function recovery than those with low tubular TRPA1 expression (OR = 7.14; 95%CI 1.35–37.75; *p* = 0.02). High TRPA1 expression was independently associated with nontotal recovery of renal function (adjusted OR = 6.86; 95%CI 1.26–37.27; *p* = 0.03). Conclusion: High tubular TRPA1 expression was associated with the nontotal recovery of renal function. Further mechanistic studies are warranted.

## 1. Introduction

Acute kidney injury (AKI) is characterized by a sharp decline in the glomerular filtration rate and manifests as azotemia [1,2]. A large portion of patients with severe complications of AKI requires renal replacement therapy [3]. AKI also results in serious health burdens because of its association with high morbidity and mortality [4]. Patients with AKI are at risk of chronic kidney disease (CKD). Over the years, most severe CKD eventually proceeds to end-stage renal disease (ESRD) [5,6,7]. If available, immediate treatment of AKI would not only reduce morbidity and mortality, but also subsequent CKD.

Acute tubular necrosis (ATN), including renal tubular cell damage and death, is the most common cause of AKI in hospitalized patients. ATN can be precipitated by acute ischemic or toxic event or sepsis [8]. Oxidative stress plays a crucial role in the pathophysiology of ATN [9]. Oxidative stress characterized by increases in reactive oxygen species (ROS) and/or reactive nitrogen species after an insult to the kidneys can initiate a complex mechanism that directly or indirectly leads to tubular injury [10,11]. However, a valid antioxidant treatment for AKI remains lacking [12]. 

Transient receptor potential ankyrin 1 (TRPA1) is a nonselective transmembrane cation channel involving Ca^2+^ permeability, which can be activated by toxic or inflammatory mediators, such as ROS [13]. Previous studies reported that TRPA1 in neurons acts as a gatekeeper of inflammation [14]. Recent studies have shown that TRPA1 is expressed in various types of non-neuronal cells, including renal tubular cells [15]. Activation of TRPA1 in these non-neuronal cells may aggravate the inflammatory response [16,17]. However, two experimental animal studies suggested that TRPA1 protects against sepsis or angiotensin-II induced kidney injury [18,19]. Consequently, the role of renal TRPA1 in AKI is not exactly known. 

The present study identified the association between renal tubular TRPA1 expression with oxidative stress, which is an activator of TRPA1, and the severity of renal injury in patients with ATN. It also investigated the association of tubular TRPA1 expression with total recovery of renal function and mortality. 

## 2. Materials and Methods

### 2.1. Study Design and Participants

We retrospectively enrolled 52 adult inpatients with AKI and biopsy-proven ATN at Changhua Christian Hospital on 1 January 2000. The biopsy-proven ATN patients who meet the criteria of Acute Kidney Injury Network (AKIN) and were aged ≥18 years were included. The AKI inpatients admitted due to obstructive etiologies (as determined by renal ultrasound), chronic dialysis patients, kidney transplant recipients, and patients with active malignancy were excluded. Each patient was followed up for three months so that renal recovery from AKI could be assessed. Six patients who underwent follow-up for less than three months were excluded; hence, 46 patients were finally selected for further investigation. In addition, six patients with normal renal function and no other remarkable comorbidities underwent nephrectomy for localized circumscribed tumors and the uninvolved poles of their removed kidneys were regarded as normal renal tissues. The study was approved by the Institutional Review Board of Changhua Christian Hospital (approval number 150912). Written informed consent was obtained from all subjects.

Renal function was measured during follow-up visits until total recovery of estimated glomerular filtration rate (eGFR), death, or the end of follow-up. The endpoint was the total (return to baseline eGFR, within a 10% margin of error) recovery of eGFR within three months following AKI and mortality. Baseline renal function was determined from the last available serum creatinine value within one year before hospitalization or the lowest inpatient serum creatinine value after AKI if outpatient serum creatinine value was unavailable. 

Demographic data, including gender, age, comorbidities, and medications, as well as urine protein excretion rate measured by the urine protein-to-creatinine ratio, were recorded at the time of AKI. Heart failure included the diagnoses of congestive or systolic heart failure, diastolic heart failure, or cardiomyopathy based on the manual review of medical charts before or at the time of AKI. The diagnosis of diabetes mellitus was based on the American Diabetes Association criteria, and hypertension was dependent on medical history and/or the use of antihypertensive medication.

### 2.2. Laboratory Data 

Serum levels of hemoglobin, creatinine, albumin, total cholesterol, triglyceride, uric acid, sodium, and potassium and urine levels of creatinine and protein were measured in accordance with standardized procedures at the Department of Laboratory Medicine, Changhua Christian Hospital. eGFR, calculated using the Chronic Kidney Disease Epidemiology Collaboration (CKD-EPI) formula, was utilized to evaluate renal function.

### 2.3. Immunohistochemistry (IHC)

Formalin-fixed, paraffin-embedded renal tissue sections (4 µm) were placed on coated slides, dewaxed with xylene, and rehydrated in serial dilutions of alcohol, followed by washing with phosphate buffered saline solution. Activity of endogenous peroxidase was blocked by incubation in 3% H_2_O_2_. Antigen retrieval was performed by boiling in 10 mM citrate buffer for 20 min. The slides were washed three times with PBS after incubation with rabbit polyclonal anti-TRPA1 antibodies (Alomone Labs., Jerusalen, Israel) at 1:2000 dilution and mouse monoclonal 8-hydroxy-2′-deoxyguanosine antibodies (ab48508, Abcam, Cambridge, MA, USA) at 1:500 dilution for 30 min at room temperature, respectively. The reaction was visualized using the polymer-based MACH4 DAB Detection Kit (Biocare Medical, Concord, CA, USA) in accordance with the manufacturer’s instructions, and the slides were incubated with horseradish peroxidase/Fab polymer conjugate for another 30 min. Finally, peroxidase activity was visualized by incubation with 3,3′-diaminobenzidine tetrahydrochloride (DAB) as the substrate for 5 min and hematoxylin as the counterstain.

Computer-assisted quantitative analysis was performed as previously described. In brief, we randomly selected at least five glomeruli and 10 nonoverlapping high-power fields for each renal cortical section and captured images by Olympus Microscope BX51 (Olympus, Tokyo, Japan) equipped with a digital color camera (DP21; Olympus, Tokyo, Japan). The captured images were then analyzed using Image Pro-Plus software (Version 6.0; Media Cybernetics, Silver Spring, MD, USA). Quantitative immunohistochemical staining value was calculated as the integrated optical density divided by the total area occupied by the DAB-stained and hematoxylin-stained cells of each slide [20].

### 2.4. Histopathology

Formalin-fixed, paraffin-embedded renal tissues including ATN and normal control were sectioned at 4 µm thickness and stained for histological examination. These sections were stained with a periodic acid-Schiff staining kit (Merck Millipore, Billerica, MA, USA) and Masson’s trichrome Kit (American Master Tech Scientific, Lodi, CA, USA) to determine the severity of tubular injury and percentage of interstitial fibrosis, respectively. All sections were examined by a pathologist (T.-C.S.) unaware of the clinical and laboratory data. The characteristics of tubular injury included tubular cell swelling, loss of brush border, or nuclear condensation. The severity of tubular injury was scored from 0 to 4 according to the percentage of the injured area of the section (0—no change; 1—changes affecting 1–25%; 2—changes affecting 25–50%; 3—changes affecting 50–75%; 4—changes affecting 75–100% of the section).

### 2.5. Statistical Analysis

Results are expressed as a percentage, median (interquartile range, IQR), or mean ± standard deviation. Kolmogorov–Smirnov test was utilized for all variables to test normal distribution. Non-normally distributed variables were analyzed by nonparametric statistical tests. Mann–Whitney U test and Pearson’s chi-squared test or Fisher’s exact test were performed to compare two groups for continuous and categorical variables, respectively. We performed univariate logistic regression analysis to calculate the crude of odds ratio (OR) of nonrecovery of total renal function or death within three months after ATN for all variables. Subsequently, multivariate logistic regression analysis was performed to calculate the adjusted OR for age, sex, and each variable. We calculated the cumulative incidences of mortality and total recovery of renal function during the follow-up period by using the Kaplan–Meier method and compared the results between the high and low TRPA1 expression groups by using the log-rank test. All statistical analyses were performed using SAS 9.4 (SAS Institute Inc., Cary, NC, USA). Statistical significance was considered at *p* < 0.05 in two-tailed tests. 

## 3. Results

### 3.1. Demographic and Clinical Characteristics of Patients

Fifty-two patients with biopsy-proven ATN were enrolled in the retrospective cohort study. Of the 52 patients, six were excluded because of follow-up less than three months. No patients started dialysis at the time of kidney biopsy. During the follow-up period, 12 patients (26.09%) completely recovered renal function. Among the 34 patients (73.91%) without complete recovery of renal function, 10 patients (21.74%) died, as seen in Figure 1. Table 1 shows the baseline demographic, laboratory data, and renal histopathology of the ATN patients. These patients are divided into patients with complete recovery of renal function (recovery group, *n* = 12) and those without complete recovery of renal function (nonrecovery or death group, *n* = 34). Patients of both groups were similar in age; gender distribution; presence of diabetic mellitus, hypertension, and heart failure; severity of AKI; levels of serum albumin, cholesterol, triglyceride, uric acid, sodium, and potassium; scores of tubular injury and interstitial inflammation; percentage of interstitial fibrosis; use of angiotensin-converting-enzyme inhibitors or angiotensin-II receptor blockers; and immunosuppressive treatment. Compared with the nonrecovery group, the complete recovery group had lower baseline serum creatinine level, higher baseline eGFR and hemoglobin levels, and lower percentage of tubular atrophy in the renal interstitium (all *p* < 0.05).

### 3.2. Association of Tubular Expression of TRPA1 with Expression of 8-OHdG or Tubular Injury Score Among Patients with ATN and Normal Subjects

The expression of renal TRPA1 on renal biopsy specimen was significantly higher in the patients with ATN than in the normal controls, as seen in Figure 2A. These ATN patients with high expression of renal TRPA1 had higher expression of renal 8-OHdG than those with low expression of renal TRPA1, as seen in Figure 2A,B (*p* = 0.033). Moreover, the patients with ATN and high renal TRPA1 expression had severe tubular injury according to the tubular injury scoring scale compared with those with low renal TRPA1 expression, as seen in Figure 2A,C (*p* = 0.006).

### 3.3. Association of Tubular TRPA1 Expression with Complete Recovery of Renal Function

Our patients were divided into two groups according to renal tubular TRPA1 expression: those with high (*n* = 22) and low (*n* = 24) expression of renal tubular TRPA1. Kaplan–Meier analysis revealed a higher incidence of complete recovery of renal function during the three-month follow-up in the low TRPA1 expression group than in the high tubular TRPA1 expression group (*p* = 0.02), as seen in Figure 3. In univariable and age- and sex-adjusted logistical regression analysis, as seen in Table 2, high tubular TRPA1 expression remained significantly associated with noncomplete recovery of renal function during the three-month follow-up (*p* = 0.02, *p* = 0.03, respectively). Compared with the AKI patients with low tubular TRPA1 expression, the OR for noncomplete recovery of renal function during the three-month follow-up was 7.14 (95%CI 1.35–37.75) in the AKI patients with high tubular TRPA1 expression. After adjustment for age and gender, high expression of tubular TRPA1 remained a significant risk factor for noncomplete recovery of renal function during the three-month follow-up (adjusted OR 6.86; 95%CI 1.26–37.27). In addition to the high expression of TRPA1, univariable and age- and sex-adjusted logistical regression analysis found that high tubular atrophy, low baseline eGFR, and low level of hemoglobin were also significantly associated with noncomplete recovery of renal function during the three-month follow-up (all *p* < 0.05). 

### 3.4. Association of Tubular TRPA1 Expression with Mortality

Kaplan–Meier analysis revealed a higher incidence trend of mortality in ATN patients with high tubular TRPA1 expression during the three-month follow-up than in those with low tubular TRPA1 expression (*p* = 0.07), as seen in Figure 4. 

## 4. Discussion

In this clinical observational study, TRPA1 was upregulated in the renal tubules of patients with ATN. In these patients with ATN, the tubular expression of TRPA1, a redox-sensing Ca^2+^-influx channel [21], is positively associated with 8-hydroxydeoxyguanosine, a marker of oxidative DNA damage and oxidative stress [22]. We also have demonstrated the positive correlation of TRPA1 expression level with the severity of tubular injury. 

The generation of oxidative stress after AKI is a major determinant of AKI; however, the effects of AKI on the renal redox system remains elusive [23]. TRPA1, an oxidative stress-sensitive Ca^2+^-permeable channel, can be activated by endogenous inflammatory agents produced on oxidative stress, such as H_2_O_2_, 4-hydroxynonenal, 4-oxononenal, and cyclopentenone prostaglandin 15-deoxy-delta (12,14)-prostaglandin J (2) (15d-PGJ(2)) [24,25]. Therefore, the positive correlation between TRPA1 expression and oxidative stress is expected. 

TRPA1 is an oxidative sensor and gatekeeper for inflammation. However, the role of TRPA1 in tissue inflammation and injury remains controversial. Some studies demonstrated that TRPA1 promotes inflammation and tissue injury in neurons or non-neuronal cells [13,17,26,27,28,29]. By contrast, a few studies suggested that TRPA1 exerts antioxidative, anti-inflammatory, organ-protective effects [30,31]. Literature with regard to TRPA1 and AKI is limited. A recent experimental animal study has suggested that TRPA1 plays a protective role in Ang II-induced renal injury possibly by inhibiting macrophage-mediated inflammation [19]. Another experimental animal study demonstrated TRPA1 may protect against sepsis-induced kidney injury by modulating mitochondrial biogenesis and mitophagy [18]. However, our previous study showed that renal tubular epithelial TRPA1 may act as an oxidative stress sensor to mediate ischemia-reperfusion-induced kidney injury through mitogen-activated protein kinases (MAPKs) and nuclear factor-*k*B (NF-*k*B) signaling. Thus, the role of TRPA1 in renal injury warrants further investigation.

In the present study, the AKI patients with high tubular TRPA1 expression had severe tubular injury. The result suggests that high TRPA1 expression in renal tubules may be a risk factor of tubular injury in AKI patients. However, the corresponding clinical role of renal tubular TRPA1 after AKI remains elusive. Therefore, we further investigated the association between clinical outcomes in AKI patients with ATN and TRPA1 expression in renal tubules.

The incidence of complete recovery of renal function was low in AKI patients with high expression of renal tubular TRPA1, and the patients with high expression of renal tubular TRPA1 had high odds of nonrecovery of renal function. This result suggests TRPA1 is associated with the progression of AKI to CKD. Progression of chronic kidney disease after acute kidney injury has a strong effect on long-term mortality [32]. As expected, the incidence of mortality in AKI patients with high TRPA1 expression was high because these patients had poor renal outcomes following AKI, although the result did not achieve statistical significance (*p* = 0.07) due to low case numbers.

The present study has several limitations. First, clinically, renal biopsy is not routinely performed in AKI patients, especially in AKI patients whose causes of AKI are known. Therefore, our results do not represent the association of TRPA1 with ATN in the total AKI population. Second, the relatively small sample size in the study lessens the statistical power of the results. Third, compared with prospective studies, retrospective cohort studies have lower statistical quality because of some unmeasured confounders. Fourth, although tubular 8-OHdG is an oxidative marker, it is not a direct activator of renal tubular TRPA1. Conversely, 4-hydroxy-2-nonenal (4-HNE) is an oxidative marker and a direct activator of renal tubular TRPA1 and thus requires further investigation to confirm the conclusion drawn from 8-OHdG staining. Fifth, this retrospective cohort study is correlational research, and thus cannot comprehensively elaborate on the causality of different expression levels of tubular TRPA1, tubular injury, and renal outcome. Therefore, the association of tubular TRPA1 expression with renal function or histopathology or clinical renal outcome of the different TRPA1 expression levels may be attributed to the severity of ATN. The role of tubular TRPA1 in AKI and its participatory mechanism in AKI remain to be elucidated. Further large prospective clinical studies or basic studies are warranted to investigate the biological role of TRPA1 in renal tubular injury after AKI. 

In conclusion, high tubular TRPA1 expression was associated with a low probability of renal recovery in patients with ATN. High tubular TRPA1 expression was associated with the severity of tubular injury and poor renal outcomes following AKI. These findings suggest that tubular TRPA1 is a potential therapeutic target for AKI. The mechanism of TRPA1 in different AKI models warrants further investigation to confirm the roles of TRPA1 in AKI. 

## 5. Conclusions

High renal tubular TRPA1 expression in AKI patients with biopsy-proven ATN was associated with the nontotal recovery of renal function.

## Figures and Tables

**Figure 1 jcm-08-02187-f001:**
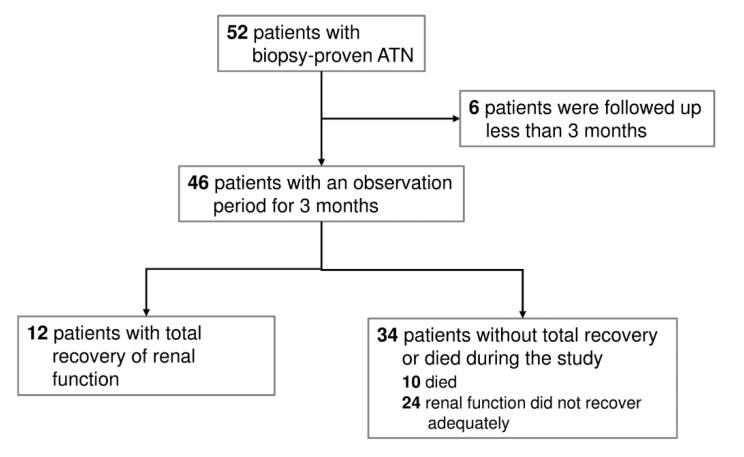
Flowchart presenting the selected biopsy-proven acute tubular necrosis (ATN) population.

**Figure 2 jcm-08-02187-f002:**
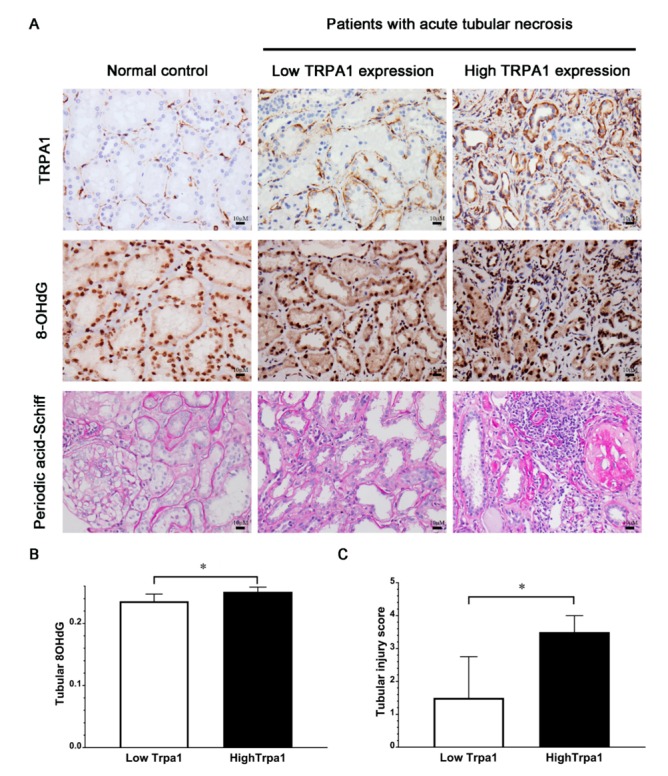
Different staining of kidney tissues from patients with ATN and association of TRPA1 expression with oxidative stress or tubular injury score. (**A**) Representative images of immunohistochemical staining of TRPA1, 8-OHdG, and periodic acid-Schiff staining of kidney tissues from patients with ATN and normal controls; 8-OHdG, an oxidative stress marker (**B**) QISV of tubular 8-OHdG (**C**) Tubular injury score. ATN patients were stratified into high and low expression groups by the cutoff value of 0.194 for tubular TRPA1 QISV based on the ROC curve analysis. Data are expressed as mean ± SD. * *p* < 0.05; TRPA1—Transient receptor potential ankyrin 1; 8-OHdG—8-hydroxy-2’-deoxyguanosine; QISV—quantitative immunohistochemical staining value; ROC—receiver operating characteristic; SD—standard deviation.

**Figure 3 jcm-08-02187-f003:**
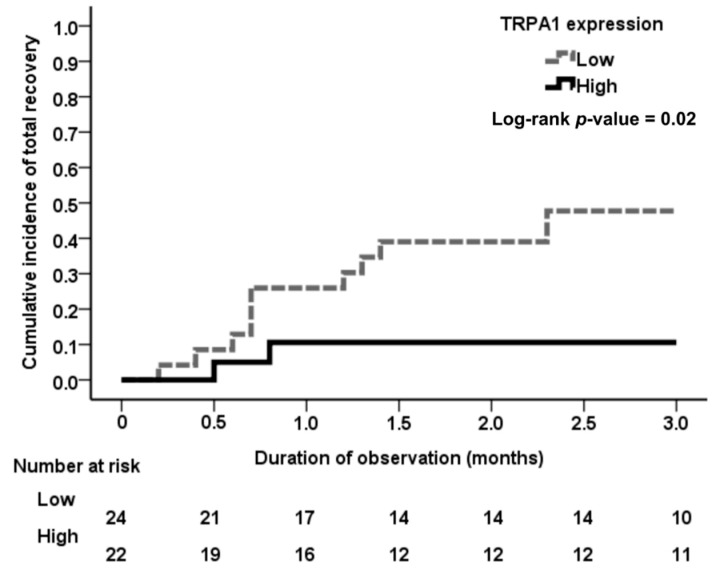
Cumulative incidence of total recovery of renal function among the ATN patients with different expression levels of tubular TRPA1. Incidence rate of the events of total recovery of renal function was significantly higher in the low tubular TRPA1 expression group than in the high tubular TRPA1 expression group during the follow-up period (log-rank test; *p* = 0.02).

**Figure 4 jcm-08-02187-f004:**
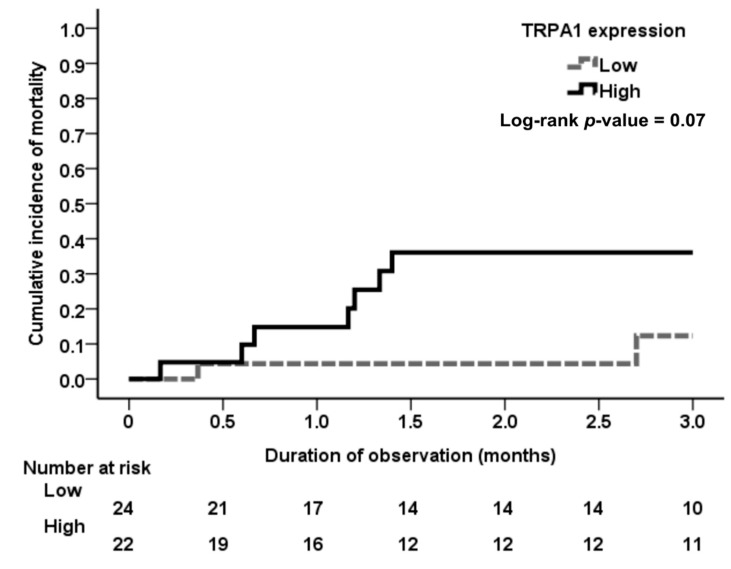
Cumulative incidence of mortality among the ATN patients with different expression levels of tubular TRPA1. Although ATN patients with high expression of tubular TRPA1 had a higher incidence of all-cause mortality than those with low expression of tubular TRPA1 during the follow-up period, the result was not statistically significant (log-rank test; *p* = 0.07). The severity of acute kidney injury may play a mediating role in all-cause mortality. Therefore, further research excluding the mediating factor is warranted.

**Table 1 jcm-08-02187-t001:** Baseline demographic and laboratory data and renal histopathology of acute tubular necrosis patients with and without total recovery of renal function within three months.

Characteristics	Total Recovery (*n* = 12)	Nonrecovery or Death ^a^ (*n* = 34)	*p* ^b^
**Demographics**			
Age (years)	46.2 ± 21.7	56.8 ± 17.8	0.15 ^b^
Male (*n* (%))	8 (66.7%)	21 (61.8%)	0.76 ^c^
Diabetes mellitus (*n* (%))	1 (8.3%)	13 (38.2%)	0.05 ^d^
Hypertension (*n* (%))	2 (16.7%)	10 (29.4%)	0.33 ^d^
Heart failure (*n* (%))	0 (0%)	3 (8.8%)	0.39 ^d^
Severity of AKI	3 (25%)	8 (23.5%)	0.60 ^d^
AKIN stage I (*n* (%))	9 (75%)	26 (76.5%)	
AKIN stage II or III (*n* (%))	46.2 ± 21.7	56.8 ± 17.8	
**Laboratory data**			
Baseline serum creatinine (mg/dL)	1.0 (0.8–1.2)	1.5 (0.9–2.7)	0.03 ^b^
Baseline eGFR (CKD-EPI) (mL/min/1.73m^2^)	88.7 (64.7–113.5)	47.7 (20.7–87.5)	0.004 ^b^
Urinary PCR (mg/g)	96.4 (30.0–976.0)	661.5 (100.0–5432.0)	0.05 ^b^
Hemoglobin (g/dL)	11.7 (9.1–13.4)	9.6 (8.7–10.8)	0.03 ^b^
Serum albumin (g/dL)	2.6 (1.8–3.2)	2.8 (2.2–3.3)	0.57 ^b^
Serum cholesterol (mg/dL)	131 (119.0–260)	193 (157–252)	0.33 ^b^
Serum triglyceride (mg/dL)	197.4 ± 132.6	165.6 ± 94.6	0.53 ^b^
Serum uric acid (mg/dL)	8.6 (7.7–13.4)	8.4 (6.6–9.6)	0.44 ^b^
Serum sodium (mmol/L)	137 (133.5–140.0)	133.5 (131–140)	0.51 ^b^
Serum potassium (mmol/L)	4.4 (3.5–4.9)	3.9 (3.4–4.1)	0.15 ^b^
**Histopathology**			
Tubular injury score	2 (1–3)	2 (1–4)	0.18 ^b^
Tubular atrophy (%)	0 (0–1.5)	6 (3–10)	<0.001 ^b^
Interstitial inflammation score	1 (0–1)	1 (1–1)	0.06 ^b^
Interstitial fibrosis (%)	7.0 ± 4.9	10.4 ± 8.4	0.37 ^b^
**Medications**			
ACEI or ARB (*n* (%))	2 (16.7%)	7 (20.6%)	0.57 ^d^
Immunosuppressants (*n* (%))	2 (16.7%)	13 (38.2%)	0.16 ^d^

Data are expressed as *n* (%) for categorical data and as mean ± standard deviation or median (interquartile range) for continuous data. AKI—acute kidney injury; AKIN—Acute Kidney Injury Network; CKD-EPI—Chronic Kidney Disease Epidemiology Collaboration; eGFR—estimated glomerular filtration rate; PCR—protein-to-creatinine ratio; ACEI—angiotensin-converting-enzyme inhibitors; ARB —angiotensin II receptor blockers. ^a^ Includes partial recoveries and nonrecoveries. ^b^ Mann–Whitney U test. ^c^ Pearson’s chi-squared test. ^d^ Fisher’s exact test.

**Table 2 jcm-08-02187-t002:** Logistical regression for nonrecovery of total renal function or death within three months after acute tubular necrosis.

Variables	Univariable		Model 1 (Adjusted for Age and Sex)	
	OR (95%CI)	*p* Value	OR (95%CI)	*p* Value
High tubular TRPA1 expression	7.14 (1.35–37.75)	0.02	6.86 (1.26–37.27)	0.03
Hypertension	2.08 (0.39–11.27)	0.39	1.84 (0.33–10.28)	0.49
Diabetes mellitus	6.81 (0.78–59.10)	0.08	5.34 (0.58–49.25)	0.14
Tubular atrophy (%)	1.96 (1.16–3.32)	0.01	2.01 (1.14–3.55)	0.02
Interstitial fibrosis (%)	1.08 (0.96–1.21)	0.19	1.06 (0.95–1.19)	0.29
Baseline eGFR (mL/min/1.73 m^2^)	0.97 (0.94–0.99)	0.01	0.97 (0.95–0.99)	0.02
Urinary protein-to-creatinine ratio (10 mg/mg)	1.00 (1.00–1.01)	0.14	1.00 (1.00–1.01)	0.14
Hemoglobin (g/dL)	0.65 (0.45–0.93)	0.02	0.68 (0.47–0.99)	0.04
Concomitant use of ACEIs or ARBs	1.30 (0.23–7.32)	0.77	1.30 (0.22–7.80)	0.78
Concomitant use of immunosuppressants	3.10 (0.58–16.41)	0.18	4.41 (0.74–26.29)	0.10

OR—Odds ratio; CI—Confidence Interval; TRPA1—Transient receptor potential ankyrin 1; ACEI—Angiotensin-converting enzyme inhibitor; ARB—angiotensin-II receptor blocker; eGFR—estimated glomerular filtration rate.

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
