# Peer review of "Renal Tubular TRPA1 as a Risk Factor for Recovery of Renal Function from Acute Tubular Necrosis"

_jcm, 2019, doi:10.3390/jcm8122187_

Round 1
Reviewer 1 Report
The Authors made substantial improvements to this manuscript since the last review. The content has become much clearer and more interesting to read.
I have attached a version of the manuscript with minor suggested changes.
The only major comment remaining is the following:
How does the discovery of TRPA1's predictive power change the landscape of literature around this topic. Is TRPA1 just another pathologic marker for severity of ATN?
If not: do the authors expect to be able to easily measure this chemical in the urine or blood in the future so as to present clinical benefit?
A consideration of these questions can be presented in the discussion.

Author Response
The Authors made substantial improvements to this manuscript since the last review. The content has become much clearer and more interesting to read.
Response:
We thank the reviewer for the affirmative response to our revised manuscript.
I have attached a version of the manuscript with minor suggested changes.
Response:
We thank the reviewer for the constructive suggestions. We have revised the manuscript in accordance with the reviewer’s comments. We believe that our revised manuscript is now clearer and more concise, and that the reviewer will find the changes satisfactory.
Page 3. Line 47. them
Response:
We modified the sentence,“The retrospective cohort study recruited 46 adult patients with AKI and biopsy-proven ATN and followed up for more than 3 months, ” on Page 3, Lines 46-47 as follows:
The retrospective cohort study recruited 46 adult patients with AKI and biopsy-proven ATN and followed them up for more than 3 months.
Page 3. Line 47. The subjects were
Page 3. Line 48. comparison of
Response:
We modified the sentences,“They were divided into high and low renal tubular TRPA1 expression groups for the evaluation of the total recovery of renal function and mortality within 3 months, ” on Page 3, Lines 47-49 as follows:
The subjects were divided into high- and low-renal-tubular TRPA1-expression groups for the comparison of the total recovery of renal function and mortality within 3 months.
Page 3. Lines 53-54. This comparator group is not alluded to before. Was there a comparator group who did not develop AKI in addition to the study cohort described in the methods? Please include a sentence about how the TRPA1 levels were obtained
Response:
We thank the reviewer for the observation. We agree that the comparator group was not alluded to previously. Six patients with normal renal function and no other remarkable comorbidities and who underwent nephrectomy for localized circumscribed tumors were recognized as the comparator group, as stated in Materials and Methods, Lines 104-106.
We modified the sentence, “Tubular TRPA1 expression was significantly increased in the ATN patients than in the comparator group and positively correlated with renal oxidative stress and injury,” on Page 3, Lines 53-54 as follows:
The expression level of tubular TRPA1 was detected by quantitative analysis of the immunohistochemistry of biopsy specimens from ATN patients.
Page 3, line 56, please also add point estimates
Response:
We thank the reviewer for the advice and add point estimates as follows:
The AKI patients with high tubular TRPA1 expression showed a high incidence of non-total renal function recovery than those with low tubular TRPA1 expression (OR=7.14; 95%CI 1.35−37.75; P=0.02).
Page 3, Line 60. But this conclusion does not follow what is said in the results. High tubular TRPA1 expression is associated with severity of renal injury and with renal recovery outcomes. Where does oxidative stress come into play?
Response:
We thank the reviewer for raising this important issue. We modified the sentence, “High tubular TRPA1 expression, correlated with oxidative stress and tubular injury, was associated with the non-total recovery of renal function,” on Page 3, Lines 59-61 as follows:
High tubular TRPA1 expression was associated with the non-total recovery of renal function.
Oxidative stress act as the activator of TRPA1, as stated on Page 16, Lines 293-298, Discussion:
TRPA1, an oxidative stress-sensitive Ca2+-permeable channel, can be activated by endogenous inflammatory agents produced on oxidative stress, such as H2O2, 4-hydroxynonenal, 4-oxononenal, and cyclopentenone prostaglandin 15-deoxy-delta (12,14)-prostaglandin J (2) [15d-PGJ(2)] .Therefore, the positive correlation between TRPA1 expression and oxidative stress is expected.
Page 4, Lines 71-72. Would say something like: Over years, most severe CKD eventually proceed to ESRD.
Response:
We removed the sentence, “The progression of CKD to end-stage renal disease (ESRD) is usually inevitable once CKD develops,” on Page 4, Lines 71-72 and replaced it with the following statement:
Over the years, most severe CKD eventually proceeds to end-stage renal disease (ESRD).
Page 4, Lines 76. in hospitalized patients
Response:
We modified the sentence, “Acute tubular necrosis (ATN) including renal tubular cell damage and death is the most common cause of AKI,” on Page 4, Lines 76 as follows:
Acute tubular necrosis (ATN), including renal tubular cell damage and death, is the most common cause of AKI in hospitalized patients.
Page 4, Lines 90. is not known exactly.
Response:
We modified the sentence in Page 4, Lines 90 “Consequently, the role of renal TRPA1 in AKI should be investigated clinically.” as follows:
Consequently, the role of renal TRPA1 in AKI is not known exactly.
Page 5, Line 5. How are you measuring oxidative stress? Aren’t you looking at severity AKI only?
Response:
We thank the reviewer for raising this important issue. Immunohistochemical staining with mouse monoclonal 8-hydroxyl-2’-deoxyguanosine (8-OHdG, a marker of oxidative stress to DNA) antibodies on renal biopsy specimen from AKI patients and quantitative immunohistochemical staining were applied to evaluate the oxidative stress on kidneys. In addition to the severity of AKI, we identified the association between the expression of renal tubular TRPA1 and oxidative stress because oxidative stress materials such as ROS are TRPA1 agonists. We modified the sentence, “The present study identified the association of the expression of renal tubular TRPA1 with oxidative stress, and the severity of renal injury in patients with ATN,” on Page 5, Lines 91-92 as follows:
The present study identified the association between renal tubular TRPA1 expression with oxidative stress, which is an activator of TRPA1, and the severity of renal injury in patients with ATN.
Page 5, Line 101. How were their diagnoses determined?
Response:
We modified the sentence “Those AKI inpatients owing to prerenal and obstructive etiologies, chronic dialysis patients, kidney transplant recipients, and patients with active malignancy were excluded.” as follows:
The AKI inpatients admitted due to obstructive etiologies (as determined by renal ultrasound), chronic dialysis patients, kidney transplant recipients, and patients with active malignancy were excluded.
Page 5, Line 108. For which analysis?
Response:
We modified the sentences, “In addition, six patients with normal renal function and no other significant comorbidities underwent nephrectomy for localized circumscribed tumors were recognized as the comparator group. The uninvolved poles of their removed kidneys were regarded as normal renal tissue.” on Page 5, Lines 106-109 as follows:
In addition, six patients with normal renal function and no other remarkable comorbidities underwent nephrectomy for localized circumscribed tumors and the uninvolved poles of their removed kidneys were regarded as normal renal tissue.
Page 15. Figure 4. Did the multivariate time to outcome analysis adjust for tubular injury found on the pathology?
Response:
We thank the reviewer for raising this important issue. In Figure 4, the comparison of different TRPA1 expression levels at the time to mortality was not adjusted for the tubular injury detected on the pathology. Hence, addition sentences were included on Page 16, Lines 281-283 in accordance with the reviewer’s suggestion as follows:
The severity of AKI may play a mediating role in all-cause mortality. Therefore, further research excluding the mediating factor is warranted.
Page 16. Line 282. I would call this the mediating factor.
Response:
We have modified the sentences, “The severity of AKI may play a confounding role in all-cause mortality. Therefore, further research excluding the confounding factors is warranted,” as follows:
The severity of AKI may play a mediating role in all-cause mortality. Therefore, further research excluding the mediating factor is warranted.
Page 17, Lines 321-323. This is an awkward sentence.
Response:
We removed the sentence, “Therefore, AKI patients with high expression of renal tubular TRPA1 are at risk of CKD following AKI progression to ESRD.” on Page 17, Lines 321-323.
Page 18, Line 330. do not represent
Response:
We modified the sentence, “Therefore, our results cannot stand for the association of TRPA1 with ATN in the total AKI population,” on Page 18, Lines 330 as follows:
Therefore, our results do not represent the association of TRPA1 with ATN in the total AKI population.
Page 18, Lines 331. This was stated in the previous statement
Response:
We removed the sentence, “Clinically renal biopsy is not routinely performed for AKI,” on Page 18, Line 331.
The only major comment remaining is the following:
How does the discovery of TRPA1's predictive power change the landscape of literature around this topic. Is TRPA1 just another pathologic marker for severity of ATN? If not: do the authors expect to be able to easily measure this chemical in the urine or blood in the future so as to present clinical benefit?
A consideration of these questions can be presented in the discussion.
Response:
We thank the reviewer for raising this important issue and giving us constructive suggestions. The retrospective cohort study is a correlation research. Thus, this study cannot elaborate in-depth the causality of different expression levels of tubular TRPA1, tubular injury and renal outcome. However, our basic unpublished study was added, which is currently under review in Oxidative Medicine and Cellular Longevity, to demonstrate the detrimental role of renal tubular TRPA1 in tubular injury via an in vivo model. This unpublished study is included on the pages after the response to reviewer 1.
No urinary or blood assay is available for TRPA1 because this protein is not secreted into the blood nor urine. Hence, we should comprehensively investigate the mechanistic roles of tubular TRPA1 in AKI and combine available urinary biomarkers to broaden their applications.
We modified the sentence, “Fourth, the association of tubular TRPA1 expression with renal function or histopathology or clinical renal outcome for the different expression levels of TRPA1 may be attributed to renal tubular injury,” in the Discussion section, Lines 334-337 as follows:
Fourth, this retrospective cohort study is a correlation research, and thus cannot comprehensively elaborate on the causality of different expression levels of tubular TRPA1, tubular injury, and renal outcome. Therefore, the association of tubular TRPA1 expression with renal function or histopathology or clinical renal outcome of the different TRPA1 expression levels may be attributed to the severity of ATN
Reference:
Unpublished in vivo data was attached as follows:
Our in vivo study showed that trpa1-/- mice exhibited reduced levels of renal ischemia-reperfusion induced inflammation and injury in the kidney compared with WT mice. The levels were identified on the basis of an alleviation of increased indices of oxidative stress, inflammation, dysfunction, and injury in the kidney (Figures 3 and 4).
Figure 3. Renal tubular injury induced by renal ischemia-reperfusion injury (IR) is lessened in trpa1-/- mice. (A) Representative images of periodic acid-Schiff (PAS) staining in renal sections obtained from two genotypes of mice with two different treatments. The magnification of each panel was 400×. (B) Tubular injury score was calculated according to the percentage of injured area of tubular cross sections. The scores system is described in the methods. Data in each group are mean ± SEM from 6 mice. ∗P < 0.05 versus the sham group in each genotype; #P < 0.05 versus the IR group of WT mice.
Figure 4. Increases in biomarker levels of renal oxidative stress, inflammation, dysfunction and injury induced by renal ischemia-reperfusion injury (IR) are all alleviated in trpa1-/- mice. (A) Quantitative data of chemiluminescence (CL) counts using renal tissues for analyses. Data are presented as CL counts per 10 sec/µg protein from renal tissue lysates. (B) MCP-1, monocyte chemoattractant protein 1. (C) MIP-2, macrophage inflammatory protein 2, (D) BUN, blood urea nitrogen, (E) blood level of creatinine, (F) tissue levels of NGAL (neutrophil gelatinase-associated lipocalin), a biomarker of acute kidney injury, were analysed by ELISA. Data in each group are mean ± SEM from 6 independent experiments. ∗P < 0.05 versus the sham group in each genotype; #P < 0.05 versus the IR group of WT mice.
Summary of Figures 3 and 4 is provided as follows: Results revealed that TRPA1 play a role in aggregating renal inflammation and injury after ischemia-reperfusion insult to kidney.
After ischemia-reperfusion injury, a kind of mouse model of AKI |
WT mice |
TRPA1-/- mice |
Tubular injury score |
High |
Less |
MCP-1 (inflammatory chemokines) |
High |
Less |
MIP-2 (inflammatory chemokines) |
High |
Less |
BUN and serum creatinine ( renal function) |
High |
Less |
NGAL ( a biomarker of AKI) |
High |
Less |
Reviewer 2 Report
Wu et al reported that tubular TRPA1 expression was significantly increased in the ATN patients than in the comparator group and the AKI patients with higher tubular TRPA1 expression had a high incidence of non-total renal function recovery. The study is interesting. However, to be accepted for publication, the following need to be done.
Major concerns:
Histological analysis and grading need to be done to evaluate TRPA1 expression in the kidney tissue and its co-relations with the typical ATN markers need to be statistically analyzed. authors could also make a ki-square test to divide patients into 4 groups (TRPA1 high and low; recover and non-recover), to see if TRPA1 levels are important to ATN recovery. What is the level of urinary cystatin c, KIM1, and oxidative stress markers (NGAL or MDA)? Those are related to tubular damage and need to be checked. What is the level of urinary inflammation or oxidative stress markers in ATN patients before and after recovery? Authors need to show strong evidence that TRPA1 expression levels are related to the progression of CKD. again, a ki-square test needs to be done.
Author Response
Wu et al reported that tubular TRPA1 expression was significantly increased in the ATN patients than in the comparator group and the AKI patients with higher tubular TRPA1 expression had a high incidence of non-total renal function recovery. The study is interesting. However, to be accepted for publication, the following need to be done.
Response:
We thank the reviewer for the affirmative response to our revised manuscript. We will revise the manuscript in in accordance with the following comments and suggestions.
Histological analysis and grading need to be done to evaluate TRPA1 expression in the kidney tissue and its co-relations with the typical ATN markers need to be statistically analyzed.
Response:
We thank the reviewer for raising these issues. The expression levels of tubular 8-OHdG and TRPA1 in kidney tissues from ATN patients were detected by quantitative analysis of immunohistochemistry with rabbit polyclonal anti-TRPA1 antibodies and mouse monoclonal 8-OHdG antibodies, respectively. Tubular 8-OHdG is regarded as a marker of oxidative stress to DNA, and oxidative stress materials are also activators of TRPA1. The severity of tubular injury in kidney tissues from ATN patients were determined by the tubular injury score, which ranges from 0 to 4 based on the percentage of the injured area of a section. ATN patients with high expression levels of renal tubular TRPA1 exhibited a remarkable more serious tubular injury (Figure 2) compared with those with low expression levels of renal tubular TRPA1. However, our clinic study is a retrospective cohort investigation conducted between January 1, 2000 and December 31, 2005. During this period, no typical ATN markers were collected. TRPA1 expression in the kidney tissue and its co-relations with the typical ATN markers will be investigated in our future study.
Authors could also make a ki-square test to divide patients into 4 groups (TRPA1 high and low; recover and non-recover), to see if TRPA1 levels are important to ATN recovery.
Response:
We thank the reviewer for giving us this important advice. We have provided a cross table and divided patients into four groups in accordance with the reviewer’s suggestion (Table 1). The results revealed that AKI patients with low expression levels of TRPA1 presented a significantly high probability of total recovery of renal function. Therefore, the tubular TRPA1 expression level is vital for total renal recovery in AKI patients with biopsy-proven ATN.
Table 1.
|
Low-TRPA1-expression group |
High-TRPA1-expression group |
P-value |
|
(n=24) |
(n=22) |
|
Outcome |
|
|
|
Total Renal recovery n (%) |
10(41.7%) |
2(9.1%) |
0.012 |
Non-total Renal recovery |
14(58.3%) |
20(90.9%) |
|
Data are expressed as n (%) for categorical data
What is the level of urinary cystatin c, KIM1, and oxidative stress markers (NGAL or MDA)? Those are related to tubular damage and need to be checked.
Response:
We thank the reviewer for this important advice. Urinary tubular injury biomarkers, such as urinary cystatin C, KIM-1, and NGAL, are important for the early detection of renal injury and prediction of kidney disease progression. However, our clinic study is a retrospective cohort research conducted between January 1, 2000 and December 31, 2005. During this period, no urine samples from the ATN patients were collected and preserved. Therefore, it’s a pity that we cannot measure these urinary biomarkers. Nevertheless, our future prospective clinical study will include these important urinary biomarkers.
What is the level of urinary inflammation or oxidative stress markers in ATN patients before and after recovery?
Response:
We thank the reviewer for this important advice. However, our clinic study is a retrospective cohort investigation conducted between January 1, 2000 and December 31, 2005. During this period, no urine samples from the ATN patients were collected and preserved. Therefore, we cannot compare the differences between urinary inflammation and oxidative stress markers in ATN patients before and after recovery.
Authors need to show strong evidence that TRPA1 expression levels are related to the progression of CKD. again, a ki-square test needs to be done.
Response:
We thank the reviewer for this important advice. The conclusion of our revised manuscript is that high tubular TRPA1 expression is independently associated with the non-total recovery of renal function in patients with AKI and biopsy-proven ATN. The role of tubular TRPA1 expression in CKD progression was not investigated in our study. Therefore, we removed the sentence, “Tubular TRPA1 may involve in chronic kidney disease following AKI...,” in the Conclusion section of the abstract in the first version of the manuscript following the reviewers’ suggestion to avoid awkwardness and confusion among readers.
Round 2
Reviewer 2 Report
It is a pity that authors can not provide more urinary ROS biomarkers or compare the differences between ATN patients before and after recovery.
However, a single immunostaining is not enough to confirm ROS status. Since the examination are mainly done by IHC staining on kidney biopsies, other oxidative stress markers could be further checked by IHC. 4-HNE (4-hydroxy-2-noneal, a marker for lipid peroxidation) amd nitrotyrosine (a marker for protein oxidation) immunostaining needs to be done and to be quantified between patient groups to confirm the conclusion made from 8-OHdG staining.
Further information can be found from the following paper [Detecting Reactive Oxygen Species by Immunohistochemistry Geou-Yarh Liou and Peter Storz]
Author Response
Response to the Reviewer’s Comments
We would like to thank the reviewer for the comprehensive assessments, constructive criticisms, and comments that are invaluable for improving the manuscript. We have revised the manuscript in accordance with the comments and suggestions. We believe that the revised manuscript is now clear and concise and the reviewer will find that the corresponding changes are satisfactory.
Response to the reviewer #2:
It is a pity that authors cannot provide more urinary ROS biomarkers or compare the differences between ATN patients before and after recovery.
Response:
We thank the reviewer for understanding the difficulties of our study. We will collect and preserve urine samples from the ATN patients to detect urinary ROS biomarkers. We will then determine the differences in these biomarkers between pre- and post-recovery ATN patients in future clinical study in accordance with the reviewer’s suggestion.
However, a single immunostaining is not enough to confirm ROS status. Since the examination are mainly done by IHC staining on kidney biopsies, other oxidative stress markers could be further checked by IHC. 4-HNE (4-hydroxy-2-noneal, a marker for lipid peroxidation) and nitrotyrosine (a marker for protein oxidation) immunostaining needs to be done and to be quantified between patient groups to confirm the conclusion made from 8-OHdG staining.
Response:
We thank the reviewer for this constructive criticism and totally agree with the reviewer’s suggestion. Indeed, oxidative stress markers of kidney specimens should further be checked through immunohistochemical staining. Among such markers, 4-HNE is particularly important because it represents the level of oxidative stress in kidneys and serves as a direct activator of the TRPA1 channel. Accordingly, we added some sentences to page 18 (line 333) in the Discussion section as shown below
Fourth, although tubular 8-OHdG is an oxidative marker, it is not a direct activator of renal tubular TRPA1. Conversely, 4-hydroxy-2-nonenal (4-HNE) is an oxidative marker and a direct activator of renal tubular TRPA1 and thus requires further investigation to confirm the conclusion drawn from 8-OHdG staining.
Further information can be found from the following paper [Detecting Reactive Oxygen Species by Immunohistochemistry Geou-Yarh Liou and Peter Storz]
Response:
We thank the reviewer for providing the vital manuscript “Detecting Reactive Oxygen Species by Immunohistochemistry” published in Methods in Molecular Biology by Geou-Yarh Liou and Peter Storz. Indeed, the detailed protocol can help us determine altered oxidative-stress levels in renal tubules by detecting ROS-induced alterations in macromolecules through immunohistochemistry.

This manuscript is a resubmission of an earlier submission. The following is a list of the peer review reports and author responses from that submission.
Round 1
Reviewer 1 Report
Wu et al. observed a different expression levels of TRPA1 in ATN patients, and it is positively correlated with renal oxidative stress and injury. The AKI patients with higher TRPA1 expression had high incidence of non-total recovery of renal function. The authors suggested that tubular TRPA1 may involve in chronic kidney disease following AKI. The study is interesting but one needs more evidence to make such conclusion.
Major concerns:
Authors devided ATN patients into two groups in Fig 3 and 4 (TRPA1 high and low), what are the baseline and laboratory data analysis of these two groups. are the creatinine and eGFR different between these two groups? or could TRPA1 level is related with the level of creatinine and eGFR? authors could also make a ki-square test to devide patients into 4 groups (TRPA1 high and low; recover and non-recover), to see if TRPA1 levels are important to ATN recovery.
What is the diagnosis of ATN? Diagnosis can be made by FENa. Authors should provide these data. What is the level of urinary cystatin c, KIM1, and oxidative stress markers (NGAL or MDA)? Those are related to tubular damage and need to be done.
As mentioned, TRPA1 is a gatekeeper of inflammation. What is the link between TRPA1 level and kidney inflammation? Authors need provide more evidence showing that whether kidney inflammation was different between TRPA1 high and low patients.
In Fig. 2 A, TRPA1 seems to localize in kidney interstatial cells in both control patients and TRPA1 low patients? However, in the high group, it is mainly in the tubules. Pictures with higher quality need to be provided, if the localization of TRPA1 changes.
Again, in Fig. 2, 8-OHdG is a marker of oxidative stress. From the figure, one would say the the control group have much higher 8-OHdG intensity than the others. Authors need to check the staining pattern and replace with proper ones. One also needs to stainings for 4-HNE and NGAL, to confirm the state of oxidative stress in such patients.
What is the level of urinary inflammation or oxidative stress markers in ATN patients before and after recovery? Authors need to show strong evidence that TRPA1 expression levels are related with the progression of CKD.
Minor concerns: in Fig. 2, scale bars are missing.
Reviewer 2 Report
The manuscript submitted by Chung-Kuan Wu et al. examines the association of biopsy derived tissue TRPA1 in cases of ATN as compared to controls gotten from nephrectomies. They were able to show an association between the higher levels of TRPA1 and non-recovery of renal function. This finding may be novel and add to our tools to better diagnose AKI outcomes.
There are two major issues with respect to this manuscript that need to be addressed. They are outlined below:
1)what is added to the broader literature with this finding? how useful are these findings. Kidney biopsies are not performed routinely for the diagnosis of ATN and thus how could this marker be used in the future? Even if this easy assays for this marker could be developed, how does it add to the already available array of urinary biomarkers?
2) the manuscript does not present the goals of the study well. At no point in the manuscript is it clear that the authors are testing the association between tissue levels of TRPA1 and outcomes. The manuscript should be rewritten with a clear hypothesis, clear methods and clear results of those methods discussed. The material is likely all there: but needs to be stated more clearly.
For more detailed suggestions I have included the manuscript with my comments.
If addressed well, the manuscript has merit.
